# Pectin methylesterase activity is required for RALF1 peptide signalling output

Ann-Kathrin Rößling[1,2], Kai Dünser[1,3], Chenlu Liu[1,2], Susan Lauw[4,5], Marta Rodriguez-Franco[6], Lothar Kalmbach[1], Elke Barbez[1,2]*, Jürgen Kleine-Vehn[1,2,3]*

[1]Institute of Biology II, Molecular Plant Physiology (MoPP), University of Freiburg, Freiburg, Germany; [2]Center for Integrative Biological Signalling Studies (CIBSS), University of Freiburg, Freiburg, Germany; [3]Institute of Molecular Plant Biology (IMPB), Department of Applied Genetics and Cell Biology, University of Natural Resources and Life Sciences (BOKU), Vienna, Austria; [4]Core Facility Signalling Factory & Robotics, University of Freiburg, Freiburg im Breisgau, Germany; [5]Centre for Biological Signalling Studies (BIOSS), University of Freiburg, Freiburg im Breisgau, Germany; [6]Institute of Biology II, Cell Biology, University of Freiburg, Freiburg im Breisgau, Germany

**\*For correspondence:**
elke.barbez@cibss.uni-freiburg.de (EB);
juergen.kleine-vehn@biologie.uni-freiburg.de (JK-V)

**Abstract** The extracellular matrix plays an integrative role in cellular responses in plants, but its contribution to the signalling of extracellular ligands largely remains to be explored. Rapid alkalinisation factors (RALFs) are extracellular peptide hormones that play pivotal roles in various physiological processes. Here, we address a crucial connection between the de-methylesterification machinery of the cell wall component pectin and RALF1 activity. Pectin is a polysaccharide, contributing to the structural integrity of the cell wall. Our data illustrate that the pharmacological and genetic interference with pectin methyl esterases (PMEs) abolishes RALF1-induced root growth repression. Our data suggest that positively charged RALF1 peptides bind negatively charged, de-methylesterified pectin with high avidity. We illustrate that the RALF1 association with de-methylesterified pectin is required for its FERONIA-dependent perception, contributing to the control of the extracellular matrix and the regulation of plasma membrane dynamics. Notably, this mode of action is independent of the FER-dependent extracellular matrix sensing mechanism provided by FER interaction with the leucine-rich repeat extensin (LRX) proteins. We propose that the methylation status of pectin acts as a contextualizing signalling scaffold for RALF peptides, linking extracellular matrix dynamics to peptide hormone-mediated responses.

## eLife assessment

This **fundamental** study provides **convincing** evidence for pectin modification as a requirement for RALF peptide signalling altering the apoplastic pH, adding further support for a key role of RALF peptides in linking the assembly and dynamics of the extracellular matrix with cellular activity and function. Data that have been added in comparison to a previous version have enhanced the study. The study should be of interest to anyone studying signaling and specifically to plant cell biologists.

## Introduction

Plants perceive various external signals, but how signals interact with the extracellular matrix is poorly understood. Pectin is a heteropolysaccharide, primarily composed of galacturonic acid units, contributing to the mechanical strength and integrity of the cell wall. Pectin (homogalacturonan) plays a vital

role in growth control with an emerging role in signalling (*Wolf, 2022*). In the cell wall, PMEs catalyse the removal of methyl groups from the initially esterified pectin, giving rise to negatively charged carboxylic acid groups and altering the interaction of pectin with various extracellular components (*Peaucelle et al., 2008*). Here, we address how the methylation status of pectin in the extracellular matrix contributes to the integration of RALFs, which are extracellular cysteine-rich plant peptide hormones. RALF peptides are involved in multiple physiological and developmental processes, ranging from organ and pollen tube growth to the modulation of immune responses (*Stegmann et al., 2017*; *Abarca et al., 2021*; *Mecchia et al., 2017*; *Geldner et al., 2009*). *Catharanthus roseus* receptor-like kinase 1-like (CrRLK1L) are transmembrane proteins with an extracellular domain, consisting of two adjacent malectin-like domains, for RALF signal perception and a cytoplasmic kinase domain for intracellular signal transduction (*Escobar-Restrepo et al., 2007*). Feronia (FER) is currently the best characterized CrRLK1L, contributing to the above-mentioned developmental programs as well as the integration of environmental cues (*Feng et al., 2018*; *Gigli Bisceglia et al., 2022*). On a cellular level, RALF binding to FER leads to rapid apoplastic (extracellular) alkalinisation (*Haruta et al., 2014*), which modulates cell wall properties, ion fluxes, and enzymatic activities, thereby influencing cellular expansion rates (*Haruta et al., 2014*; *Barbez et al., 2017*; *Dünser et al., 2019*; *Dünser and Kleine-Vehn, 2015*). Besides its role in apoplastic pH, CrRLK1Ls play a role in cell wall sensing (*Hématy et al., 2007*; *Höfte, 2015*; *Shih et al., 2014*) and can bind to extracellular pectin (*Feng et al., 2018*; *Lin et al., 2022*; *Liu et al., 2024*). The FER-dependent cell wall sensing mechanism involves the direct interaction with the extracellular leucine-rich repeat extensinsLRXs in roots (*Dünser et al., 2019*), suggesting that LRXs provide a physical link between FER and the cell wall (*Herger et al., 2019*). On the other hand, root, as well as pollen-expressed LRX proteins, also bind to RALF peptides (*Mecchia et al., 2017*; *Dünser et al., 2019*; *Moussu et al., 2020*), but its contribution to mechanochemical sensing and/or growth control is largely unknown in roots. Here, we show that the PME-dependent de-methylesterification status of pectin defines the FER-dependent perception of RALF1, contributing to extracellular regulation of pH, cell wall and plasma membrane remodelling, control of receptor endocytosis as well as organ growth in roots. Our data suggest that negatively charged, de-methylesterified pectin binds to positively charged RALF1 peptides with low affinity but high avidity. Interference with the de-methylesterification of pectin, out-titrating RALF1 with small charged, de-methylesterified pectin fragments, as well as the disruption of the positive charges in RALF1, abolishes the RALF1 activity in roots. We accordingly propose that the RALF interaction with de-methylesterified pectin at the root surface is crucial for its FER-dependent perception in roots. Even though root-expressed LRX proteins are strictly required for the FER-dependent mechano-sensing in roots, the LRX proteins are not essential for RALF perception in roots (*Dünser et al., 2019*). We moreover show here that LRX proteins do not contribute to the integration of RALF-pectin signalling in roots. In conclusion, we report on the crucial contribution of pectin de-methylesterification, for FER-dependent RALF peptide hormone signalling in root growth control. Our work proposes that pectin acts as a context-dependent extracellular signalling scaffold, contributing to signalling dynamics at the plasma membrane.

## Results and discussion

The application of RALF1 peptides induces strong repression of main root growth in *Arabidopsis thaliana* (*Haruta et al., 2014*; *Figure 1A and B*), which we initially used here as a bioassay to visualise RALF1 activity. To assess if the de-methylesterification status of pectin may impact peptide signalling, we pharmacologically interfered with the de-methylesterification of pectin, using epigallocatechin gallate (EGCG). EGCG is a natural inhibitor of PME activity (*Lewis et al., 2008*) and its application lowered main root growth but its co-treatment markedly interfered with the RALF1-induced reduction in root growth (*Figure 1A and B*). In contrast, the structurally similar epicatechin (EC) is not an inhibitor of PMEs (*Lewis et al., 2008*) and did not block RALF1-induced root growth repression (*Figure 1C and D*). In agreement, EGCG but not EC application reduced the labelling of de-methyl esterified pectin in roots (*Figure 1E and F*) as visualised by the fluorescent probe COS[488] (*Mravec et al., 2014*). This set of data suggests that the EGCG-induced interference with PME activity correlates with reduced RALF1 responses in roots. To consolidate this assumption, we employed the overexpression of PECTIN METHYLESTERASE INHIBITOR 3 (*PMEI3*), which is well characterised to interfere with PME activity (*Peaucelle et al., 2008*; *Xu et al., 2022*) and accordingly also reduced the extracellular de-methylesterification of pectin in epidermal root cells (*Figure 1G and H*). In agreement with the

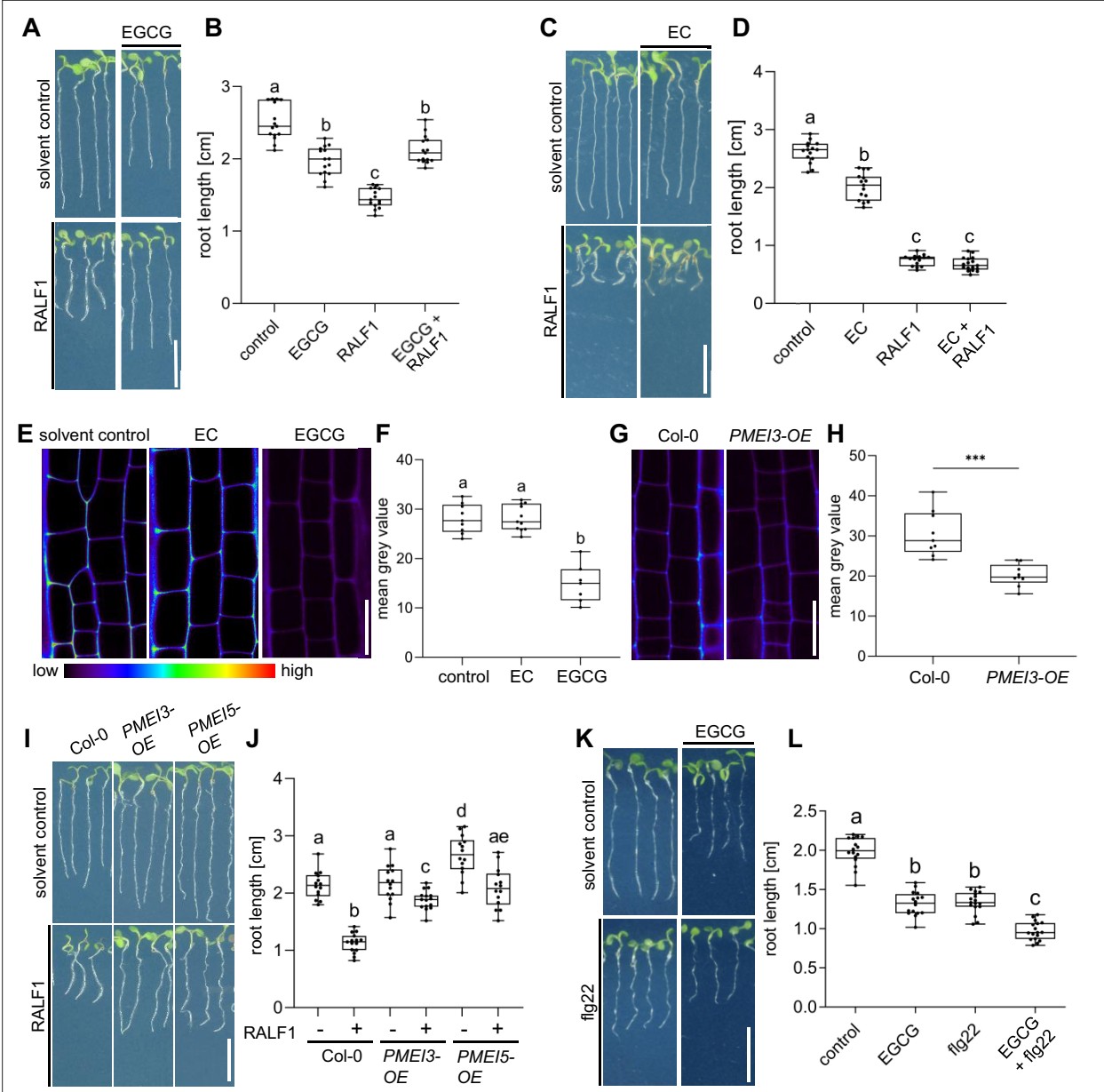

**Figure 1.** Pectin methyl esterase (PME) activity is required for rapid alkalinisation factor1 (RALF1)-induced root growth repression. (**A, B**) Three-day-old wild-type seedlings were subjected for 3 days to 1 μM RALF1 and/or 2.5 μM epigallocatechin gallate (EGCG) (**A**). (**B**) Boxplots depict the root length of wild-type seedlings under different treatments shown in (**A**). (**C, D**) Three-day-old wild-type seedlings were transferred for 3 days to solvent control, 1 μM RALF1 and/or 15 μM epicatechin (EC) (**C**). (**D**) Boxplots depict the root length of wild-type seedlings under different treatments shown in (**C**). (**E, F**) Confocal microscopy images of the root epidermal cells of 6-day-old wild-type seedlings after 3 hr treatment in liquid medium with 50 μM EC/EGCG or solvent control. Seedlings were stained with COS[488] to visualize de-methylesterified pectin. (**F**) Boxplots displaying the probe staining signal intensity are shown in (**E**). Scale bar = 25 μm, n=8–10 roots per treatment with a total number of 79–96 quantified cells. (**G, H**) Confocal microscopy images of the root epidermal cells of 6-day-old wild-type and *PMEI3-OE* seedlings, stained with COS[488] to visualize de-methylesterified pectin. (**H**) Boxplots displaying the probe staining signal intensity are shown in (**G**). Scale bar = 25 μm, n=9–11 roots per treatment with 71–77 quantified cells. (**I–J**) Three-day-old wild-type, *PMEI3-OE,* and *PMEI5-OE* seedlings were exposed for three days to 1 μM RALF1 or solvent control (**I**). (**J**) Boxplots depict the root length of wild-type compared to *PMEI3-OE* seedlings of the treatments shown in (**I**). (**K, L**) Three-day-old wild-type seedlings were exposed for three days with 0.5 μM flg22 and 15 μM EGCG or solvent control (**K**). (**L**) Boxplots depict the root length of wild-type seedlings under different treatments shown in (**K**). Statistical significance was determined by a one-way ANOVA with a Tukey Post Hoc multiple comparisons test (p<0.05, letters indicate significance categories) (**B, D, F, J, and L**) and a student´s t-test (***p=0.0001) (**H**). Boxplots: Box limits represent the 25th and 75th percentile, and the horizontal line represents the median. Whiskers display min. to max. values. Representative experiments are shown. (**A, C, I, and K**) Scale bar = 1 cm, n=11–13 roots per treatment/line.

*Figure 1 continued on next page*

Figure 1 continued

The online version of this article includes the following source data for figure 1:

**Source data 1.** Data for *Figure 1A, B*.

**Source data 2.** Data for *Figure 1C, D*.

**Source data 3.** Data for *Figure 1E, F*.

**Source data 4.** Data for *Figure 1G, H*.

**Source data 5.** Data for *Figure 1I, J*.

**Source data 6.** Data for *Figure 1K, L*.

EGCG application, *PMEI3*, as well as *PMEI5,* gain-of-function reduced the RALF1-induced root growth repression (*Figure 1I and J*).

We accordingly conclude that pharmacological and genetic interference with PME activity leads to the repression of RALF1 effects on root growth. To test the specificity of this effect, we subsequently used the peptide flagellin22 (flg22). The flg22 peptide is derived from the N-terminus of bacterial flagellin and is known to elicit root growth repression via the innate immune receptor flagellin sensitive2 (FLS2) (*Chinchilla et al., 2007*). EGCG treatments were not able to suppress, but instead additively enhanced the FLG22-induced root growth repression (*Figure 1K and L*), suggesting a rather specific effect of PME inhibition on balancing RALF1 activity.

Next, we addressed how PME activity affects RALF1-dependent root growth control. FER-dependent perception of RALF peptides may affect cell wall integrity and it is hence conceivable that the pharmacological and genetic interference with extracellular PME activity could counterbalance some extracellular effects of RALF1 signalling. Using electron microscopy, we indeed observed a RALF1-induced effect on cell wall integrity, showing swollen cell walls, but also revealed plasma membrane invaginations (*Figure 2A and B*). These pronounced RALF effects were, to our knowledge, never reported and hence we confirmed these effects independently, using live cell confocal imaging. RALF1-induced invaginations of plasma membrane markers, such as Low-Temperature inducible protein 6b (LTi6b) (*Figure 2C*) and Novel Plant SNARE 12 (NPSN12) (*Figure 2—figure supplement 1A*), were readily detectable. Similarly, confocal imaging of propidium iodide (PI)-stained cell walls of wild-type seedlings also depicted the RALF1-induced swelling of the apoplastic signal and revealed additional ectopic accumulations (*Figure 2D and F*). These RALF1-induced alterations were absent in PI-stained *fer-4* loss-of-function mutants (*Figure 2E and F*), suggesting that FER activation is required for these cellular effects of RALF1. The pharmacological (*Figure 2G and H*) as well as genetic (*Figure 2J and K*) interference with PME activity abolished the RALF1-induced alterations in cell wall labelling, phenocopying *fer-4* mutants.

PME activity could hence either indirectly counterbalance RALF1 effects on the cell wall and/or it could directly define RALF1 signalling output in root cells. Accordingly, we next tested the RALF1 impact on apoplastic pH regulation, which is a primary readout of RALF1 perception by FER (*Haruta et al., 2014*). We used 8-hydroxypyrene-1,3,6-trisulfonic acid trisodium salt (HPTS) as a fluorescent pH indicator for assessing apoplastic pH in epidermal root cells (*Barbez et al., 2017*). As expected, the application of RALF1 peptides induced an extracellular pH alkalisation in wild-type epidermal root cells (*Barbez et al., 2017*; *Figure 3A and B*). In contrast, EGCG application (*Figure 3A and B*) as well as the overexpression of PMEI3 (*Figure 3C and D*), abolished the RALF-induced alkalisation of the apoplastic pH. We accordingly conclude that interference with PME activity disrupts the primary output signalling of RALF1 in root cells.

Our data suggest that PME inhibition interferes with RALF1 signalling output, ultimately affecting cell wall properties and root growth. To assess if PME activity affects the extracellular perception of RALF1, we next investigated the ligand-induced cellular dynamics of its receptor FER. An early event of receptor-mediated signalling often involves the ligand-induced endocytosis of the receptor as has been previously demonstrated for the RALF1-receptor FER (*Yu et al., 2020*). Therefore, we next tested whether the RALF1 ligand-induced endocytosis of FER is altered upon changes in the methylation status of pectin. In agreement with previous reports, we observed the RALF1-induced internalisation of functional *FER::FER-GFP* in epidermal root cells (*Figure 3F and G*). On the other hand, the ligand-induced internalisation of FER-GFP was abolished when we pharmacologically or genetically

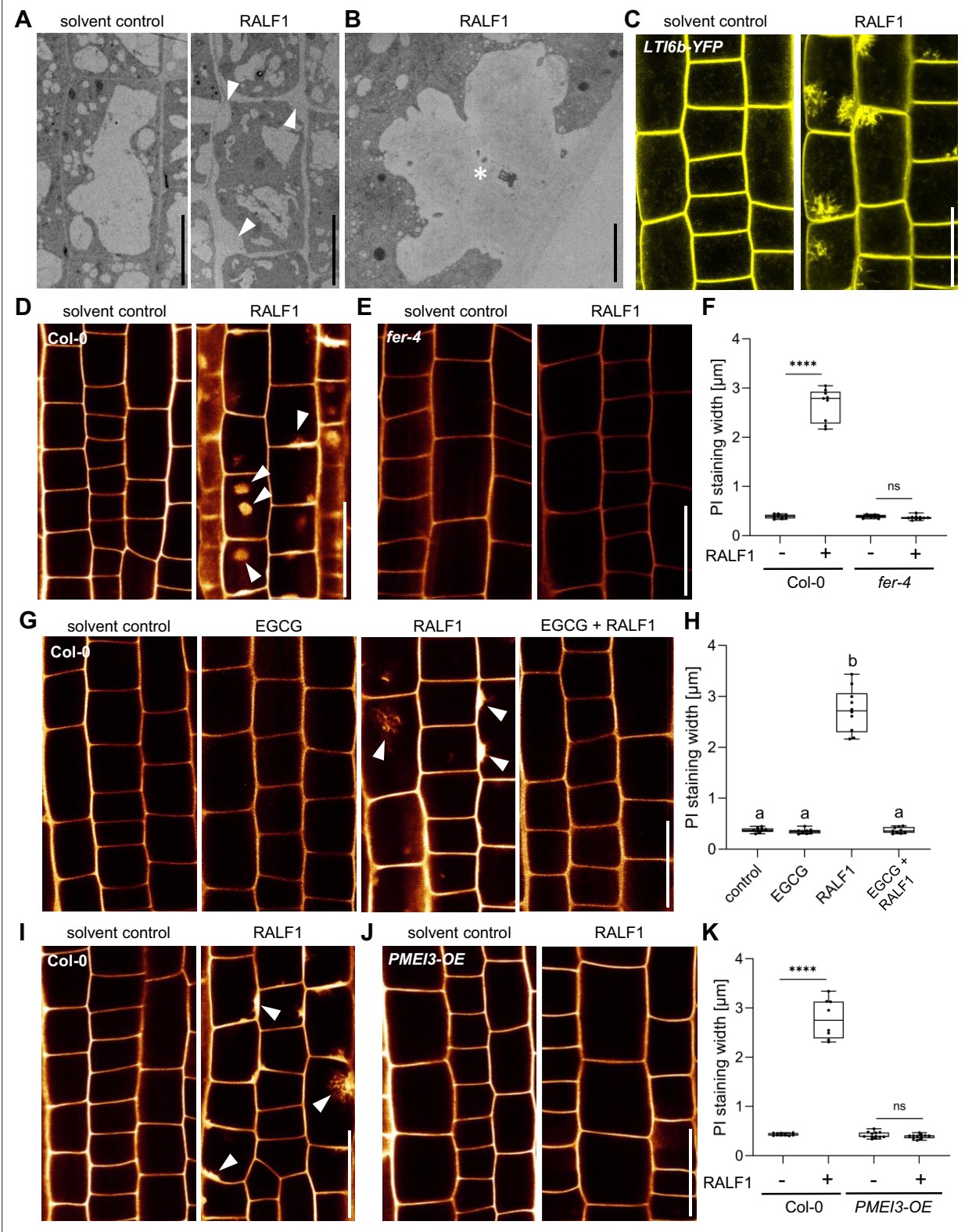

**Figure 2.** Rapid alkalinisation factor1 (RALF1) requires pectin methyl esterase (PME) activity to affect cell wall integrity. (**A, B**) Representative transmission electron microscopy (TEM) images of epidermal root cells treated for 3 hr with solvent control and 1µM RALF1, respectively. Arrowheads indicate RALF1-induced plasma membrane invaginations. Scale bar = 10 µm. Panel (**B**) shows details, Scale bar = 2 µm. (**C**) Confocal microscopy images of the root epidermal cells of 6-day-old LTI6b-YFP expressing seedlings, treated for 3 hr with solvent control or 1µM RALF1. Scale bar = 25 µm. (**D–F**) Confocal microscopy images of the root epidermal cells of 6-day-old wild-type (**D**) and *fer-4* mutant (**E**) seedlings, treated for 3 hr with 1 µM RALF1 or

*Figure 2 continued on next page*

*Figure 2 continued*

solvent control. Seedlings were mounted in propidium iodide to visualize the cell walls. Arrowheads indicate RALF1-induced alterations of the cell wall stain. (**F**) Graphs represent the width of propidium iodide (PI) signal per root under different treatments shown in (**D, E**). Scale bar = 25 µm, n=9–11 roots per treatment with a total number of 49–51 quantified cells. (**G, H**) Confocal microscopy images (**G**) and quantification (**H**) of 6-day-old roots of wild-type seedlings, treated in liquid medium with 1 µM RALF1 and/or 50 µM epigallocatechin gallate (EGCG) as well as solvent control for 3 hr. Seedlings were mounted in propidium iodide to visualize the cell walls. Arrowheads indicate cell wall invaginations. Scale bar = 25 µm, n=8–10 roots per treatment with a total number of 44–49 quantified cells. (**I–K**) Confocal microscopy images (**I, J**) and quantification (**K**) of 6-day-old wild-type (**I**) and *PMEI3-OE* (**J**) roots, treated in liquid medium with 1 µM RALF1 or solvent control for 3 hr. Seedlings were mounted in propidium iodide to visualize the cell walls. Arrowheads indicate cell wall invaginations. Scale bar = 25 µm, n=8–12 roots per treatment with a total number of 46–53 quantified cells. Statistical significance was determined by a one-way ANOVA with a Tukey Post Hoc multiple comparisons test (p<0.05, letters indicate significance categories) (**H**) or a two-way ANOVA with Bonferroni Post Hoc test (****p<0.0001) (**F, K**). Boxplots: Box limits represent the 25th and 75th percentile, and the horizontal line represents the median. Whiskers display min. to max. values.

The online version of this article includes the following source data and figure supplement(s) for figure 2:

**Source data 1.** Data for *Figure 2F*.

**Source data 2.** Data for *Figure 2H*.

**Source data 3.** Data for *Figure 2K*.

**Figure supplement 1.** Cell wall invaginations after rapid alkalinisation factor1 (RALF1) treatment in a plasma membrane marker.

---

suppressed PME activity (*Figure 3F–J*). These findings propose that the PME activity is required for the earliest events of RALF1 perception by FER in roots.

To test the specificity of this observation, we used another RALF peptide called RALF34, which is a ligand of CrRLK1L theseus1 (THE1) (*Gonneau et al., 2018*). Similar to RALF1, the application of RALF34 induced the internalisation of FER-GFP, which was fully suppressed when PME activity was inhibited (*Figure 3—figure supplement 1*). Therefore, we assume that several RALF peptides display the same or similar signalling modalities.

RALF peptides are positively charged (*Abarca et al., 2021*). Hence, it is hence conceivable that the association of positively charged RALF1 peptides with de-methylesterified, negatively charged pectin is required for its FER-dependent perception. To indirectly assess if RALF1 peptides bind de-methylesterified pectin, we tested if RALF1 peptides bind fragments of de-methylesterified oligogalacturonides (OGs).

We initially assessed potential OG binding to RALF1, using the RALF1 impact on root cells as a bioassay. The addition of high levels of free, de-methylesterified OGs into the media did not visibly affect the cell wall or FER endocytosis on its own (*Figure 4—figure supplement 1A–D*). On the other hand, free OGs in the medium strongly interfered with RALF1-induced cell wall swelling and FER receptor endocytosis in roots (*Figure 4—figure supplement 1A–D*). This effect was concentration-dependent (*Figure 4A and B*), suggesting that excess of de-methylesterified OGs in the medium can bind and out-titrate RALF1 peptides and thereby limit the RALF1 activity at the root surface.

To characterise this binding in vitro, we next used biotinylated RALF1 in biolayer interferometry (BLI), which is an optical technique for measuring macromolecular interactions by analyzing interference patterns of white light reflected from the surface of a biosensor tip. The equilibrium dissociation constant (Kd) in the range of 105 nanomolar (nM) (*Figure 4C*) suggests a physiologically relevant binding affinity. Notably, we, however, observed an initial OG association to RALF1 with a constant ($K_{on}$) in the micromolar (µM) range, but the dissociation kinetics revealed a remarkably slow rate (*Figure 4C and D*). This is indicative of relatively low affinity at the individual RALF1-OG binding event level but multiple intermolecular binding events may eventually lead to a high accumulative binding strength. We hence propose that RALF1 peptides and de-methylesterified OGs undergo low affinity but high avidity binding assemblies. This finding complements recent data, indicating that the binding of RALF peptides and de-methylesterified OGs leads to phase separation in vitro (*Liu et al., 2024*).

RALF1 contains in total eight positively charged amino acids (three lysine (K) and five arginine (R)) (*Figure 4—figure supplement 2A, B*). This is reminiscent of the well-characterised de-methylesterified pectin-binding protein polygalacturonase-inhibiting protein (PGIP), which binds to de-methylesterified pectin through a positively charged binding site formed by clustered residues of lysine (K) and arginine (R) (*Spadoni et al., 2006*). Notably, the LRX8-RALF4 complex binds de-methylesterified OGs in a charge-dependent manner in pollen (*Moussu et al., 2023*). To address the positive charges in RALF1, we synthesized a RALF1-like peptide by substituting the K and R residues (*RALF1⁻ᴷᴿ*). Thereby,

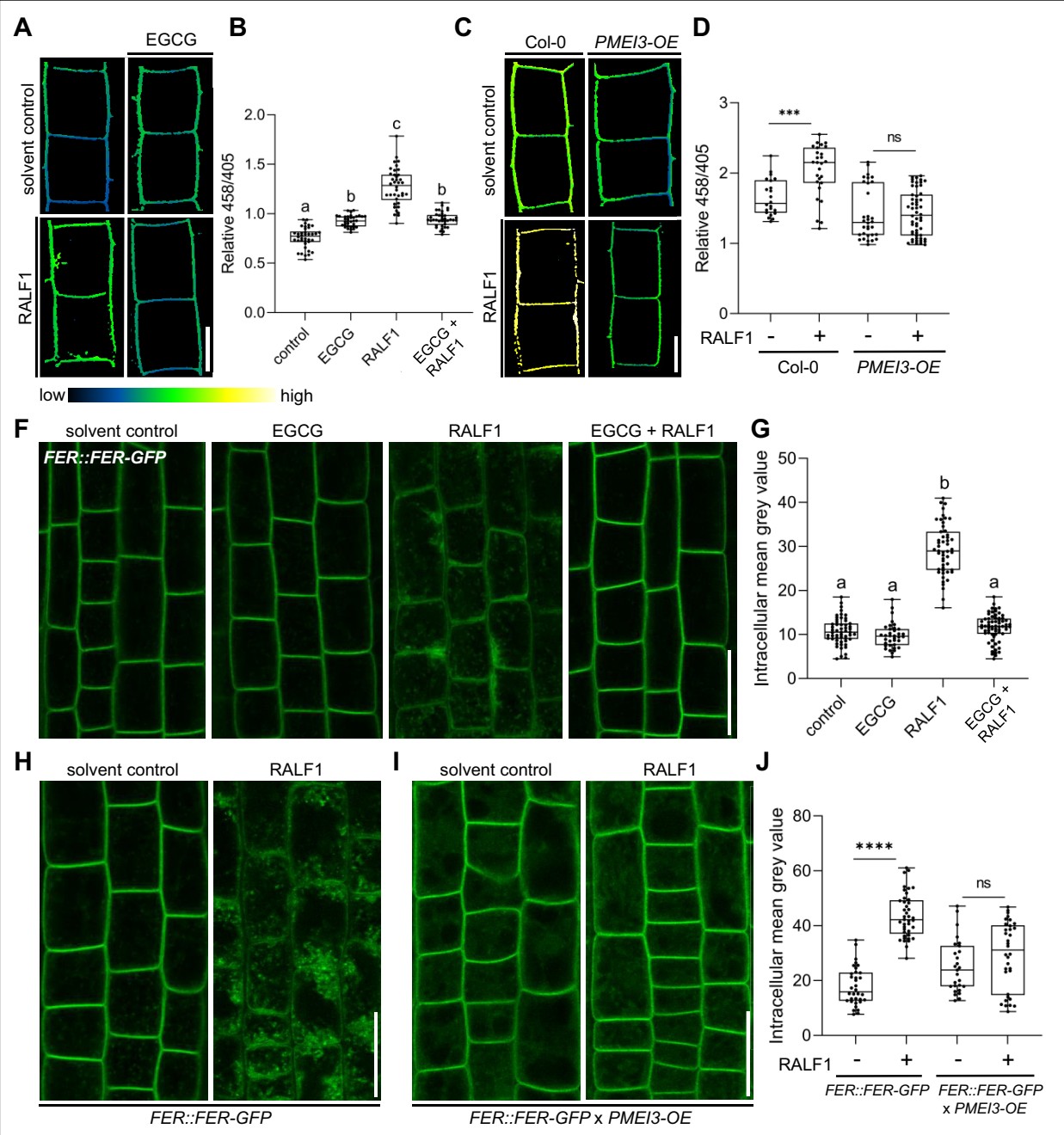

**Figure 3.** Pectin methyl esterase (PME) activity is required for rapid alkalinisation factor1 (RALF1) signalling output. (**A, B**) 8-hydroxypyrene-1,3,6-trisulfonic acid trisodium salt (HPTS)-based (458/405 ratio) pH assessment of late meristematic root cells of wild-type seedlings treated for 3 hr with 1 µM RALF1, 50 µM epigallocatechin gallate (ECGC), or both compared to solvent control. Representative confocal images (**A**) are shown. Scale bar = 10 µm. (**B**) Boxplots depict mean HPTS 458/405 intensities shown in (**A**), n=9–11. (**C, D**) HPTS-based (458/405 ratio) pH assessment of late meristematic cells in wild-type and *PMEI3-OE* seedlings treated for 3 hr with 1 µM RALF1 compared to solvent control. Scale bar = 10 µm. (**D**) Boxplots depict mean HPTS 458/405 intensities shown in (**C**), n=10–12. (**F, G**) RALF1-induced internalisation of *FER::FER-GFP* in late meristematic epidermal root cells of seedlings treated for 3 hr with 1 µM RALF1, 50 µM ECGC, or both compared to solvent control. Scale bar = 25 µm. (**G**) Boxplots displaying the mean intracellular GFP signal intensity are shown in (**F**), n=9–12. (**H–J**) RALF1-induced internalisation of feronia (FER) in late meristematic epidermal root cells in *FER::FER-GFP* (**H**) and in *FER::FER-GFP* crossed with *PMEI3-OE* (**I**) seedlings treated for 3 hr with 1µM RALF1 compared to solvent control. Scale bar = 25 µm. (**J**) Boxplots displaying the intracellular GFP signal intensity are shown in (**H, I**). Statistical significance was determined by a one-way ANOVA with a Tukey Post Hoc multiple comparisons test (p<0.05, letters indicate significance categories) (**B, G**) or a two-way ANOVA with Bonferroni Post Hoc test (****p<0.0001) (**D, J**). Boxplots: Box limits represent the 25th and 75th percentile, and the horizontal line represents the median. Whiskers display min. to max. values.

*Figure 3 continued on next page*

*Figure 3 continued*

The online version of this article includes the following source data and figure supplement(s) for figure 3:

**Source data 1.** Data for *Figure 3B*.

**Source data 2.** Data for *Figure 3D*.

**Source data 3.** Data for *Figure 3G*.

**Source data 4.** Data for *Figure 3J*.

**Figure supplement 1.** Activity of RALF34 requires pectin methyl esterase (PME) activity.

**Figure supplement 1—source data 1.** Data for *Figure 3—figure supplement 1B*.

we created a slightly negatively charged peptide in the pH range of the cell wall (*Figure 4—figure supplement 2A, C*). *RALF1$^{-KR}$* peptides are not bioactive, because they did neither affect root growth, nor cell wall integrity, nor did they induce the ligand-induced endocytosis of FER in epidermal root cells (*Figure 4—figure supplement 2D–I*). These findings suggest that the positively charged residues in RALF1 are essential for its activity in roots. Even though we cannot rule out that *RALF1$^{-KR}$* affects several characteristics of RALF1 signalling, we used this non-charged peptide to test if these residues affect the RALF1 binding to de-methylesterified pectin. Using our in vitro system, we did not observe *RALF1$^{-KR}$* binding to de-methylesterified OGs (*Figure 4—figure supplement 2J*), proposing that RALF1 and OGs indeed interact in a charge-dependent manner.

The LRX-RALF complex and its interaction with pectin exert a condensing effect on the cell wall in pollen and root hairs (*Moussu et al., 2023*; *Schoenaers et al., 2024*). Moreover, LRX proteins also bind and link FER with a cell wall-sensing mechanism in roots (*Dünser et al., 2019*). Nevertheless, LRX function in root growth appears to be independent of FER-dependent RALF signalling (*Dünser et al., 2019*). Hence, we next tested if LRX proteins are required for the joint RALF1/pectin-dependent signalling in roots. We observed that RALF1 applications still induced cell wall alterations in *lrx1 lrx2 lrx3 lrx4 lrx5* quintuple mutant roots (*Figure 5A–D*). Similar to wild-type roots (*Figure 5E and F*), pharmacological interference with PME activity still repressed RALF1 activity in the *lrx1 lrx2 lrx3 lrx4 lrx5* quintuple mutant roots (*Figure 5G and H*). These findings confirm our previous assumptions that LRX proteins are not essential for the RALF activity in roots (*Dünser et al., 2019*) and, moreover, reveal that LRX proteins are also not required for the PME-dependent control of RALF1 signalling in roots.

In conclusion, our data proposes that RALF1 peptides are associated with high avidity to de-methylesterified pectin, which is required for the RALF perception by FER. We hence conclude that the methylation status of pectin provides a conceptualising input to RALF peptide hormone signalling (*Figure 5I and J*).

Concluding remarks:

Intriguingly, the exogenous application of RALF1 does not only affect cell wall characteristics but also induces prominent plasma membrane protrusions into root epidermal cells. Plasma membrane invagination is a highly controlled process and often relates to plant endosymbiotic events (*Su et al., 2023*). The developmental importance of this RALF1 effect remains to be addressed but could pinpoint pectin signalling playing a central role in the structural coordination of cell wall and plasma membrane dynamics. Most importantly, we reveal an essential function of the cell wall component pectin for RALF1 peptide signalling (*Figure 5I and J*). We illustrate that the pharmacological and genetic interference with PME activity specifically abolishes RALF1, but not flg22 peptide, activity in roots. We conclude that PME activity is required for all hallmarks of RALF1 signalling, including the rapid alkalinisation of the cell wall and the ligand-induced endocytosis of FER. We hence propose that the generation of de-methylesterified, negatively charged pectin in the cell wall is required for the FER-dependent perception of positively charged RALF1 at the root cell surface. We show that RALF1 binds in a charge-dependent manner to de-methylesterified OGs in vitro and that the application of free OGs can out-titrate RALF1 peptides in the medium. Our in vitro data proposes low affinity but a high avidity binding of OGs and RALF1. This interaction mechanism could be key in the spatial enrichment of RALF peptides at the root surface because receptor interactions at the plasma membrane could contribute to the accumulated strength of multiple binding interactions. In agreement, after we pre-printed a previous version of this manuscript (*Rößling et al., 2023*), Moussu and colleagues proposed that the LRX8-RALF4 complex and its charge-dependent binding of de-methylesterified OGs controls extracellular condensations in pollen tubes (*Moussu et al., 2023*). The contribution of FER signalling

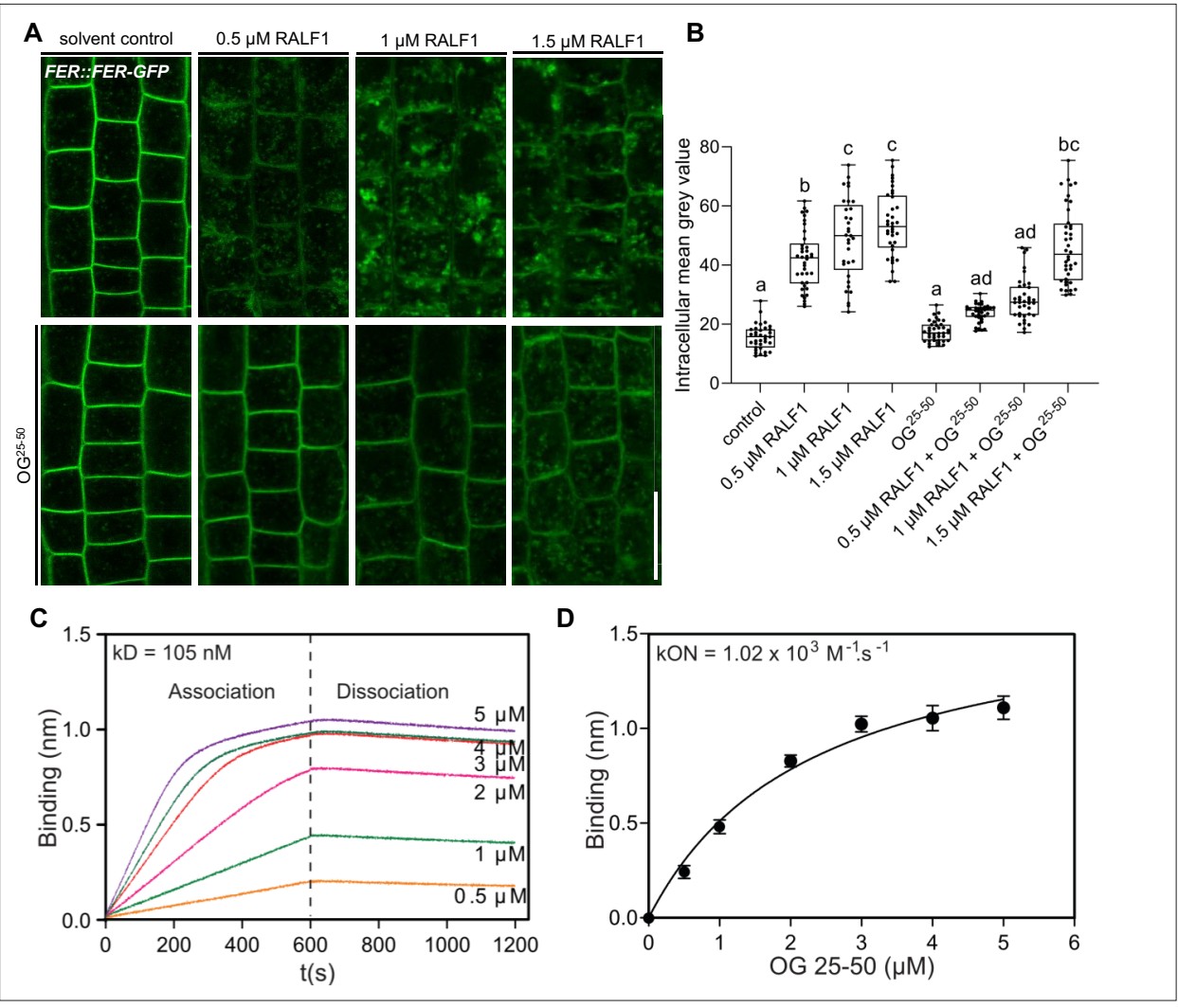

**Figure 4.** Rapid alkalinisation factor (RALF) peptides associate with de-methylesterified oligogalacturonides. (**A, B**) RALF1-induced internalisation of *FER::FER-GFP* in late meristematic epidermal root cells of 6-day-old seedlings treated for 3 hr with a concentration series of 0.5, 1, and 1.5 µM RALF1 compared to solvent control and co-treated with oligogalacturonides (OGs) with a chain length of 25–50 (OG$^{25-50}$; 50 mg/mL). (**B**) Boxplots displaying the mean intracellular GFP signal intensity are shown in (**A**). Scale bar = 25 µm, n=9–12. (**C, D**) Biolayer interferometry assays showed the binding between RALF1 and OG$^{25-50}$ with an Equilibrium dissociation constant (Kd) of 105 nM (**C**). The association and dissociation constants ($k_{ON}$ = 1.02 × 103 M-1 s-1, $k_{OFF}$ =1.07 × 10–4 s-1) imply high avidity binding (**D**). Statistical significance was determined by a one-way ANOVA with a Tukey Post Hoc multiple comparisons test (p<0.05, letters indicate significance categories) (**B**). Boxplots: Box limits represent the 25th and 75th percentile, and the horizontal line represents the median. Whiskers display min. to max. values.

The online version of this article includes the following source data and figure supplement(s) for figure 4:

**Figure supplement 1.** Application of free de-methylesterified oligogalacturonides disrupts rapid alkalinisation factor1 (RALF1) activity at the cell surface.

**Figure supplement 2.** Positive charges in rapid alkalinisation factor1 (RALF1) are required for its bioactivity.

**Source data 1.** Data for *Figure 4B*.

**Figure supplement 1—source data 1.** Data for *Figure 4—figure supplement 1B*.

**Figure supplement 1—source data 2.** Data for *Figure 4—figure supplement 1D*.

**Figure supplement 2—source data 1.** Data for *Figure 4—figure supplement 2E*.

**Figure supplement 2—source data 2.** Data for *Figure 4—figure supplement 2G*.

**Figure supplement 2—source data 3.** Data for *Figure 4—figure supplement 2I*.

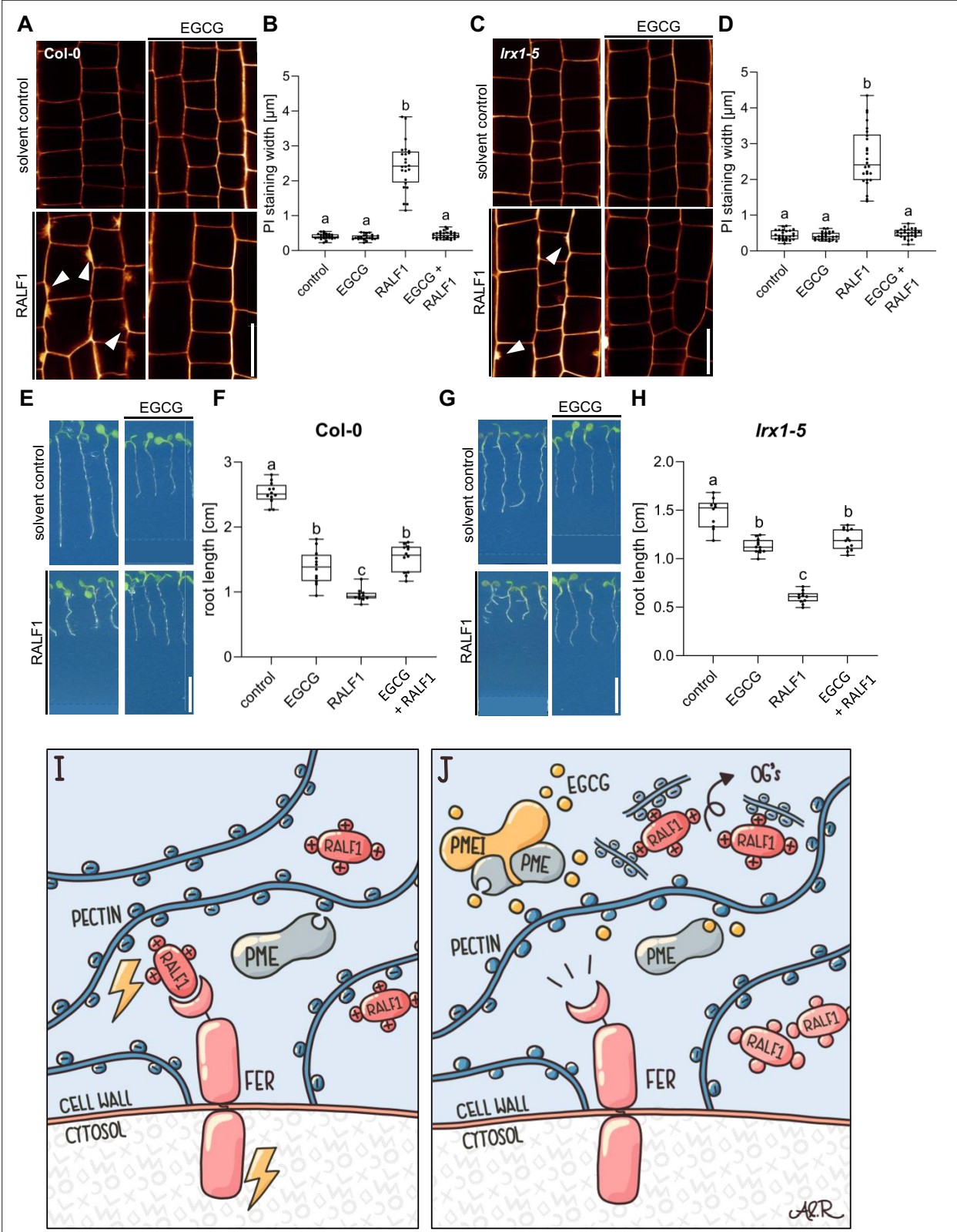

**Figure 5.** Leucine-rich repeat extensin (LRX) proteins are not essential for the pectin methyl esterase (PME)-dependent activity of rapid alkalinisation factor1 (RALF1). (**A, B**) Confocal microscopy images (**A**) and quantification (**B**) representing the average cell wall width per root under different treatments shown in (**A**), of 6-day-old roots of wild-type seedlings, treated in a liquid medium with 1 µM RALF1 and/or 50 µM EGCG as well as solvent control for 3 hr. Seedlings were mounted in propidium iodide to visualise the cell walls. Arrowheads indicate cell wall invaginations. Scale bar = 25 µm,

*Figure 5 continued on next page*

*Figure 5 continued*

n=10–12 roots per treatment with a total number of 44–52 quantified cells. (**C, D**) Confocal microscopy images (**C**) and quantification (**D**) representing the average cell wall width per root under different treatments shown in (**C**), of 6-day-old roots of *lrx1/lrx2/lrx3/lrx4/lrx5* quintuple mutant seedlings, treated in liquid medium with 1 μM RALF1 and/or 50 μM EGCG as well as solvent control for 3 hr. Seedlings were mounted in propidium iodide to visualize the cell walls. Arrowheads indicate cell wall invaginations. Scale bar = 25 μm, n=11–12 roots per treatment with a total number of 46–53 quantified cells. (**E–H**) Three-day-old wild-type (**E**) or *lrx1/lrx2/lrx3/lrx4/lrx5* quintuple mutant (**G**) seedlings transferred for three days in liquid growth medium supplemented with solvent control, 1 μM RALF1 and/or 15 μM epigallocatechin gallate (EGCG). Seedlings were transferred to a solid growth medium just before imaging. (**F, H**) Boxplots display root length of seedlings under different treatments shown in (**E, G**). Scale bar = 1 cm, n=14–16 roots per treatment/line. (**I, J**) De-methylesterified and hence negatively charged pectin is crucial for the signalling output of positively charged RALF peptides (**I**). Interference with PME activity, application of free de-methylesterified OGs, or removal of positive charges in RALF leads to abolished RALF1 output signalling (**J**). Statistical significance was determined by a one-way ANOVA with a Tukey Post Hoc multiple comparisons test ($p<0.05$, letters indicate significance categories) (**B, D, F, and H**). Boxplots: Box limits represent the 25th and 75th percentile, and the horizontal line represents the median. Whiskers display min. to max. values. Representative experiments are shown.

The online version of this article includes the following source data for figure 5:

**Source data 1.** Data for *Figure 5F*.

**Source data 2.** Data for *Figure 5H*.

remains unknown in this process, but a similar structural mechanism defines root hair growth in a FER-dependent manner (*Schoenaers et al., 2024*). Within the main root, the RALF peptide binding to de-methylesterified pectin was suggested to be sufficient to lead to its extracellular phase separation, subsequently inducing the clustering of FER receptors in the plasma membrane (*Liu et al., 2024*). It remained, however, unknown if LRX proteins are implied in this root organ growth response. LRX proteins directly interact with FER and thereby contribute to cell wall sensing in roots (*Dünser et al., 2019*). Notably, LRX proteins are not required to sense RALF1 in roots (*Dünser et al., 2019*) and also the inhibition of PMEs still represses RALF1 activity in *lrx1 lrx2 lrx3 lrx4 lrx5* quintuple mutant roots. These findings suggest an LRX-independent mode of action in the FER-RALF1-pectin signalling mechanism. Our findings (this study and *Dünser et al., 2019*) pinpoint distinct FER- and RALF-dependent roles of LRX proteins. Considering that LRX8-RALF4-pectin interaction contributes to wall integrity and pollen tube expansion (*Moussu et al., 2023*), distinct modes of cell wall integrity control may be also in place in root cells and tip-growing pollen tubes.

Based on our findings, we envision that negatively charged, de-methylesterified pectin could function as a signalling scaffold for positively charged RALF peptides, which seems crucial for its signalling via the FER receptor. It needs to be seen how precisely the RALF binding to homogalacturonan affects its interaction with FER receptors. Here it is noteworthy that the pectin sensing function of FER is physically separated from the RALF binding domain (*Gronnier et al., 2022*).

The impact of de-methylesterified pectin on RALF signalling could also lead to self-emerging feedback properties because high extracellular PME activity is expected to strongly acidify the apoplast (*Wolf et al., 2009*), which in turn would be counteracted by the de-methylesterified pectin-induced activation of RALF peptides and its consequences on apoplast alkalinisation (*Haruta et al., 2014*; *Barbez et al., 2017*; this study). The here uncovered mechanism links the pectin-related cell wall status with RALF peptide hormone signalling. A scaffold protein can tether signalling components, enhancing the efficiency and specificity of cellular signalling pathways by acting as a structural framework. In analogy, we propose that pectin functions in a PME-dependent manner as a tunable extracellular signalling scaffold.

## Materials and methods
### Plant material and growth conditions
All experiments were carried out in *Arabidopsis thaliana*, ecotype *Col-0*. The following plant lines were described in previous publications: *PMEI3-OE* (*Peaucelle et al., 2008*), *PMEI5-OE* (*Wolf et al., 2014*), *lrx1/lrx2/lrx3/lrx4/lrx5* (*Dünser et al., 2019*), *LTI6b-YFP* (*Cutler et al., 2000*), *UBQ10::NPSN12-YFP* (Wave131Y, *Geldner et al., 2009*), *FER::FER-GFP* (in *fer-4*, *Shih et al., 2014*), *fer-4* (*Shih et al., 2014*), *FER::FER-GFP* x *PMEI3-OE* (in *fer-4*, obtained by crossing). After surface sterilisation in 70% and 100% ethanol, seeds were vernalized at 4 °C for 2 days in darkness, afterward grown vertically on ½ strength

Murashige and Skoog (MS) medium plates containing 1% sucrose in a long-day regime (16 hr light and 8 hr darkness) at 21 °C.

## Chemicals and peptides

All chemicals were dissolved in dimethyl sulfoxide (DMSO) and served as solvent control, unless indicated otherwise. We used PI in a working concentration of 0.02 mg/mL for cell wall counterstaining (obtained from Sigma (MO, USA)). Epigallocatechin gallate (EGCG) was used to interfere with PME activity in liquid treatments (obtained from Cayman Chemical (MI, USA)), epicatechin (EC) was used as negative control (from Cayman Chemical (MI, USA)) and flg22 peptides as a peptide control (obtained from Sigma (MO, USA)). The RALF1 peptide (mature RALF1, with the amino acid sequence: ATTKYISY QSLKRNSVPCSRRGASYYNCQNGAQANPYSRGCSKIARCRS) and the non-charged RALF1$^{-KR}$ peptide version (amino acid sequence: ATTSYISYQSLTSNSVPCSDTGASYYNCQNGAQANPYSDGCSYIASCRS) were both synthesized by PSL GmbH (Heidelberg, Germany) and dissolved in water. The galacturonan oligosaccharides OG$^{10-15}$ and OG$^{25-50}$ were obtained from Eliceityl (Crolles, France) and Biosynth (Staad, Switzerland), respectively.

## Confocal microscopy

For image acquisition, an upright Leica TCS SP8 FALCON FLIM confocal laser scanning microscope, equipped with a Leica HC PL APO Corr 63×1.20 water immersion CS2 objective, was used. GFP was excited at 488 nm (fluorescence emission: 500–550 nm), PI at 561 nm (fluorescence emission: 640–750 nm). Experiments assessing apoplastic pH were carried out as previously described (*Barbez et al., 2017*) using 8-hydroyypyrene-1,3,6-trisulfonuc acid trisodium salt (HPTS; Sigma-Aldrich). In brief, roots of 6-day-old seedlings were treated for 3 hr in liquid ½ MS medium containing 1 µM RALF1 and/or 50 µM EGCG or the appropriate amount of solvent control (water or DMSO, respectively) for 3 hr. For the RALF1-OG titration experiment, 0.5, 1, and 1.5 µM RALF1 were mixed with OG$^{25-50}$ in a concentration of 50 mg/mL and the seedlings were incubated for 3 hr. If not mentioned otherwise, pharmacological treatments have been performed in liquid medium in multiwell plates. Prior to imaging, the roots were mounted on a solid block of ½ MS medium containing 1 mM HPTS. The block was transferred to a microscopy slide and imaged with a coverslip on top. Image processing was performed as previously described (*Barbez et al., 2017*) using a macro for Fiji. Four transversal cell walls were quantified per root. The protonated form of HPTS was excited at 405 nm and 0.5% Diode UV laser intensity, and the deprotonated form at 455 nm with 30% Argon laser intensity. Images here were obtained in sequential scan mode. Z-stacks were recorded with a step size of 420 nm, with a stack containing 20 slices on average. The Gain of the HyD detectors (or PMT for HPTS experiments) stayed constant during imaging. The pinhole was set to 111.4 µm (for HPTS experiments to 196 µm). To counterstain the cell walls, roots were mounted in PI solution (0.02 mg/mL) prior to imaging. The signal intensity of intracellular space or plasma membrane was assessed using Fiji, with four cells analyzed per root. The ROI stayed constant over the experiments and biological replicates. For quantification of the PI staining width (and, therefore, the thickness of the cell wall) the thickness of transversal and longitudinal sections stained by PI were measured using Fiji, with four to six stained plasma membranes quantified per root. For the RALF1-treated roots, only the invaginated regions were chosen for analysis.

## Synthesis of COS probes

The COS probe was generated as described in *Mravec et al., 2014*, using chitosan oligosaccharides conjugated with AlexaFluor 488 hydroxylamine (ThermoFisher, Waltham, Massachusetts, USA). In brief, 1 mg/ml oligosaccharide solution was mixed into 0.1 M sodium acetate buffer, pH 4.9. The AlexaFluor 488 (10 mg/ml in DMSO) was added to the oligosaccharide (½ of the moles of the oligosaccharide). After incubation at 37 °C for 48 hr with shaking at 1400 rpm the probe was ready. Six-day-old seedlings were transferred into liquid ½ MS medium and supplemented with DMSO, 50 µM EGCG, and EC for 3 hr, respectively. After the treatment, seedlings were staining in liquid ½ MS medium supplemented with COS$^{488}$ at a 1:500 dilution, for 30 min, and washed twice with medium. Imaging of root tips was performed directly afterward.

## Root length analysis and root growth assay

For root growth assays, seedlings were grown for 3 days on solid ½ MS plates. Afterwards, they were transferred into 3 mL of liquid ½ MS medium containing RALF1 and/or EGCG or the appropriate

amount of solvent control (water or DMSO, respectively). After 3 days of growth in liquid media with gentle agitation on a platform shaker, the seedlings were placed on solid ½ MS plates and scanned using a flatbed scanner. Root length was measured using Fiji. For statistical analyses, the GraphPad Prism Software (version 9.5.1) was used.

### Sample preparation for electron microscopy

Six-day-old wild-type seedlings were treated for 3 hr in liquid ½ MS media containing solvent control or 1 µM RALF1. After the treatment, roots were cut apart with a razor blade, immediately submerged in MTSB buffer containing 4% p-formaldehyde (PFA), and vacuum infiltrated for 15 min in a microwave oven (Pelco BioWave Pro+) at room temperature. The submerged roots were left in the fixative solution for 4 hr at room temperature and overnight at 4 °C. All further steps were performed according to *Gronnier et al., 2022*.

### In silico analysis

For assessment of the charge of the RALF1 and RALF1$^{-KR}$ peptide, the Peptide Calculator Online tool from https://www.biosynth.com/peptide-calculator was used.

### Biolayer interferometry assays

Octet@RED 96 was used to perform the binding assay between the biotinylated RALF1 or biotinylated RALF1$^{-KR}$ and OG$^{25-50}$. Both, biotinylated-RALF1 and biotinylated RALF1$^{-KR}$ (purchased from Peptide Specialty Laboratory GmbH) were solubilized in DMSO as a stock solution in a concentration of 11 mg/ml. The buffer used for the interaction was PBS pH 7.4 containing 25 µg/ml BSA. RALF1 was used as a positive control to compare the binding of RALF1$^{-KR}$ to OG$^{25-50}$. The Blank contained the biotinylated ligand without OG$^{25-50}$.

Test experiments were carried out to rule out the unspecific binding of OG$^{25-50}$ to the streptavidin biosensor and biotin. The Streptavidin biosensor was first dipped in the interaction buffer for 600 s and continued by loading with 5 µg/ml Biotinylated-RALF1 as a ligand for 600 s. The sensors were washed in the interaction buffer for 600 s followed by OG$^{25-50}$ in the concentration of 0, 0.5, 1, 2, 3, 4, and 5 µM for association (600 s). Finally, dissociation was monitored for 600 s in the interaction buffer. Three biological replicates were measured and analyzed using Octet Red Data Analysis software version 8.0.2.3. The fitting curve was selected to be a 1:1 binding model. The curves were plotted using GraphPad Prism6 software.

### Illustrations

The summary figures (*Figure 5I and J*) were created using the Concepts App for iOS (Version 6.9.2, TopHatch, Inc (Turku, Finnland)) and edited using AffinityDesigner (version 1.10, Serif (West Bridgford, UK)) by Ann-Kathrin Rößling.

## Acknowledgements

We are grateful to Niko Geldner, Gabriele Monshausen, Grégory Mouille, and Jozef Mravec for sharing published material; our team members for helpful discussions; and the LIC Imaging Center Freiburg for expertise and support. We would like to thank Rosula Hinnenberg of the EM facility at the Faculty of Biology, University of Freiburg, for her assistance with the generation of EM data. The TEM (Hitachi HT7800) and confocal laser scanning microscope (Zeiss 980) for vertical imaging were funded by DFG grants (project number 426849454 and 499026372). This work was supported by the Austrian Science Fund (FWF) (P33044 to JKV), and the German Science Fund (DFG; 470007283 to JKV and CIBSS – EXC-2189 Project ID 390939984 to JKV and to EB).

## Additional information

#### Competing interests

Jürgen Kleine-Vehn: Reviewing editor, eLife. The other authors declare that no competing interests exist.

## Funding

| Funder | Grant reference number | Author |
|---|---|---|
| Deutsche Forschungsgemeinschaft | 426849454 | Marta Rodriguez-Franco |
| Deutsche Forschungsgemeinschaft | 499026372 | Jürgen Kleine-Vehn |
| Austrian Science Fund | 10.55776/P33044 | Jürgen Kleine-Vehn |
| Deutsche Forschungsgemeinschaft | 470007283 | Jürgen Kleine-Vehn |
| Deutsche Forschungsgemeinschaft | CIBSS - EXC-2189 Project ID 390939984 | Elke Barbez, Jürgen Kleine-Vehn |

The funders had no role in study design, data collection and interpretation, or the decision to submit the work for publication.

## Author contributions

Ann-Kathrin Rößling, Conceptualization, Data curation, Validation, Investigation, Visualization, Methodology, Writing – original draft, Writing – review and editing; Kai Dünser, Conceptualization, Data curation, Formal analysis, Validation, Investigation, Methodology; Chenlu Liu, Data curation, Investigation, Methodology; Susan Lauw, Marta Rodriguez-Franco, Data curation, Investigation; Lothar Kalmbach, Supervision, Validation; Elke Barbez, Conceptualization, Supervision, Funding acquisition, Writing – original draft, Project administration, Writing – review and editing; Jürgen Kleine-Vehn, Conceptualization, Data curation, Supervision, Funding acquisition, Writing – original draft, Project administration, Writing – review and editing

## Author ORCIDs

Ann-Kathrin Rößling ⓘ https://orcid.org/0000-0002-1395-6525
Susan Lauw ⓘ https://orcid.org/0000-0002-0541-2173
Marta Rodriguez-Franco ⓘ https://orcid.org/0000-0003-1183-2075
Elke Barbez ⓘ http://orcid.org/0000-0002-5531-7916
Jürgen Kleine-Vehn ⓘ https://orcid.org/0000-0002-4354-3756

Reviewer #1 (Public review): https://doi.org/10.7554/eLife.96943.3.sa1
Reviewer #2 (Public review): https://doi.org/10.7554/eLife.96943.3.sa2
Reviewer #3 (Public review): https://doi.org/10.7554/eLife.96943.3.sa3
Author response https://doi.org/10.7554/eLife.96943.3.sa4

# Additional files

## Supplementary files
- MDAR checklist

## Data availability

All data generated or analysed during this study are included in the manuscript and supporting files.

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
