## [Editor Report · eLife assessment]

This **fundamental** study provides **convincing** evidence for pectin modification as a requirement for RALF peptide signalling altering the apoplastic pH, adding further support for a key role of RALF peptides in linking the assembly and dynamics of the extracellular matrix with cellular activity and function. Data that have been added in comparison to a previous version have enhanced the study. The study should be of interest to anyone studying signaling and specifically to plant cell biologists.

---

## [Referee Report · Reviewer #1 (Public review)]

Summary:

Rößling et al., report in this study that the perception of RALF1 by the FER receptor is mediated by the association of RALF1 with deesterified pectin, contributing to the regulation of the cell wall matrix and plasma membrane dynamics. In addition, they report that this mode of action is independent from the previously reported cell wall sensing mechanism mediated by the FER-LRX complex.

This manuscript reproduces and aligns with the results from a recently published study (Liu et al., Cell) where they also report that RALF1 can interact with deesterified pectin, forming coacervates and promoting the recruitment of LLG-FER at the membrane.

---

## [Referee Report · Reviewer #2 (Public review)]

Summary:

The study by Rößling et al. addresses the link between the biochemical constitution of the cell wall, in particular the methylesterification state of pectin with signalling induced by the extracellular RALF peptide. The work suggests that only in the presence of demethylesterifies pectin, RALF is able to trigger activation of its receptor FERONIA (FER).

Remarkably, the application of RALF peptides leads to rather dramatic FER-dependent changes in wall integrity and plasma membrane invaginations not observed before. Interestingly, RALF can be out-titrated from the wall by short pectin fragments. In addition, the study provides further evidence for multiple FER-dependent pathways by showing the presence of LRX proteins is not required for the pectin/RALF mediated signalling.

Strengths:

This work provides fundamental insight into a complex emerging pathway, or perhaps several pathways, linking pectin sensing, pectin structure and RALF/FER signalling. The study provides convincing evidence that pectin methylesterase activity is required for RALF sensing, indicating that the physical interaction of RALFs with the cell wall is important for signalling. Beyond that, the study documents very clearly how profoundly RALF signalling can affect cell wall integrity and membrane topology.

Weaknesses:

Not a weakness per se, as it cannot be avoided, but drawing conclusions from genetic material with altered pectin always suffers from the possibility of secondary effects as this cell wall component is under heavy surveillance and able to respond plastically to different cues. However, the authors take that into account and have performed adequate controls to minimize that possibility.

---

## [Referee Report · Reviewer #3 (Public review)]

In this important work, the authors show compelling evidence that the Rapid Alkalinisation Factor1 (RALF1) peptide acts as an interlink between pectin methyl esterification status and FERONIA receptor-like kinase in mediating extracellular sensing. Moreover, the RALF1-mediated pectin perception is surprisingly independent of LRX-mediated extracellular sensing in roots. The authors also show that the peptide directly binds demethylated pectin and the positively charged amino acids are required for pectin binding as well as for its physiological activity.

Some present findings are surprising; previously, the FERONIA extracellular domain was shown to bind pectin directly, and the mode of operation in the pollen tube involves the LRX8-RALF4 complex, which seems not the case for RALF1 in the present study. Although some aspects remain controversial, this work is a very valuable addition to the ongoing debate about this elusive complex regulation and signaling.

The authors drafted the manuscript well, so I do not have a lot of criticism or suggestions. The experiments are well-designed, executed, and presented, and they solidly support the authors' claims.

---

## [Author Response]

The following is the authors’ response to the original reviews.

**Reviewer #1 (Recommendations for the Authors):**
Major(a) In the study the authors focus on the RALF1 peptide. But according to expression data and the study from Abarca et al., 2021, RALF1 is not the only peptide expressed in the root and also having an impact in root growth effect. Similarly, looking at the primary sequence from RALF1 it does not differ much chemically from other RALFs such as RALF33, RALF23, RALF22, etc. So, does the cell wall pectin methylation status also have an impact on the effect of other RALFs on root growth or is that specific of RALF1?(b) In addition, is the internalization of FER depending only on RALF1 upon the methylation status of cell wall pectins? Or can other RALFs cause a similar effect potentially?(c) The authors propose that RALF1 associates with deesterifed pectin, through electrostatic interactions. To do that they perform Biolayer interferometry assays using a buffer with pH 7.4. Is that a relevant pH at the cell wall? Is possible that the authors thought that this may not change the charges of R and K residues, however, it will affect the overall charge of the peptide given the fact that it contains quite some N and Q in the exposed surface. The authors may want to consider that.(d) Moreover, the authors do not use their peptide RALF1KR, suggested as a peptide not binding OGs, as a control in their OG binding assays. That biochemical experiment should also be included to validate their results and conclusions.

We thank reviewer #1 for these comments. In this work, we focused on RALF1 but the majority of AtRALF peptides, when applied exogenously as synthetic peptides, induce RALF1like effects in *Arabidopsis* (Abarca et al., 2021; PMID: 34608971). Moreover, all RALF peptides display clusters of R and K residues and are negatively charged (Abarca et al., 2021; PMID: 34608971). In comparison to RALF1, we now also use RALF34 because it was suggested to interact also via the *Catharanthus roseus* receptor-like kinase 1-like (CrRLK1L) THESEUS1 (THE1). Notably, RALF34 also induced the internalization of FER-GFP. Moreover, the interference with PME also disrupted this activity of RALF34. Therefore, we assume that other RALF peptides display the same or similar signalling modalities. Nevertheless, it remains to be addressed if all RALF family members require PME activity.

We appreciated these comments and incorporated this aspect in the revised version of the manuscript. The pH was chosen for technical reasons associated with the used BLI buffer. As requested, we also included the RALF1-KR peptide in our OG binding assays. Under these conditions, the mutated peptides were not able to interact with the OGs anymore. Accordingly, we conclude that the K and R residues in RALF1 are crucial for its binding to demethylesterified OGs.

(e) Another important aspect is regarding their design RALF1KR mutant and its effect in planta. The authors report the following: "RALF1-KR peptides are not bioactive, because they did neither affect root growth, nor cell wall integrity, nor did they induce the ligand-induced endocytosis of FER in epidermal root cells (Figure 5D-I). These findings suggest that the positively charged residues in RALF1 are essential for its activity in roots." According to the structure published by Xiao at el. 2019, the R in the alpha helix from RALF peptides (YISYQSLKR... in RALF1 seq) is directly involved in the interaction with LLGs. So, a mutation in that R may impair the interaction of RALF1 with LLG and therefore the complex formation with FER. So, it is well possible that the effect that the authors are seeing on FER signaling and endocytosis, using this peptide variant, may not be due to the impaired capacity of the peptide to bind deesterified pectin but to not be able to be sensed by the membrane complex directly. To verify that the authors should test, either biochemically or by CoIP in planta, that their RALF1KR variant can still be perceived by the LLG-FER complex.

We agree with reviewer #1 and do not doubt that the positive charges in RALF1 likely interact with several entities. The respective sites were also covered in Liu et al., 2024 (Cell). It would be interesting to understand how the charge-dependent interaction with pectin modulates the RALF binding to the LLG-FER complex, but these experiments are beyond the scope of this manuscript. We confirmed that the negative charges in RALF1 are essential for OG binding as well as for its bioactivity. We however do not rule out that they bear additional structural functions beyond pectin binding. We clarified this aspect in the revised version. It is conceivable that the pectin and receptor complex binding of RALF1 is molecularly and mechanistically related.

(f) The authors propose in this study that this effect of RALF1-pectin mode of action on FER is independent from LRXs. That is a very interesting observation which also aligns with similar observations from other independent studies (Moussu et al., 2020; Schoenaers et al. Nat Plants, 2024; Franck et al., 2018). However, that seems to be in conflict with the previous mode of action that the authors had described in Dunser et al., 2019. In that last study the authors had described that FER constitutively interacts with LRX proteins in a direct way to sense cell wall changes. In my view the authors do not critically elaborate to explain these two contradicting results, which are key to understand the mode of action they are describing. This relevant aspect should be addressed more in depth by the authors in their discussion.

Thank you for the comment. We do not see that our findings contradict our previous work (from Dünser et al., 2019). There we concluded that LRX and FER directly interact to sense cell wall characteristics. However, the loss of *LRX* function abolished the cell wall sensing mechanism, but the respective loss-of-function and dominant negative lines were still able to detect RALF peptides. We hence proposed that the LRX/FER function is at least partially independent of the FER function in RALF perception. This is in agreement with our current study where we conclude again that FER shows LRX-dependent but also -independent modes of action.

Minor(g) In the introduction (first page), the authors write the following sentence: "RALF peptides are involved in multiple physiological and developmental processes, ranging from organ growth and pollen tube guidance to modulation of immune responses (Stegmann et al., 2017; Abarca et al., 2021)". RALFs are not involved in pollen tube guidance but pollen tube growth.So, that should be changed in the Introduction sentence. Also, in addition, the authors could cite additional references here to support the sentence such as Mecchia et al., 2017 or Ge et al. , 2017, in addition.

Thank you for pointing this out and we apologize for our flaw. We corrected the statement in the revised version of the manuscript and added the citations as requested.

(h) The new study of Schoenaers et al. Nat Plants, 2024 should now be included in the revised version.

Thank you. We implemented this reference in the revised manuscript.

**Reviewer #2 (Public Review):**
The genetic material used by the authors to strengthen the connection of RALF signalling andPME activity might not be as suitable as an acute inhibition of PME activity. The PMEI3ox line generated by Peaucelle et al., 2008 is alcohol-inducible. Was expression of the PMEI induced during the experiments? As ethanol inducible systems can be rather leaky, it would not be surprising if PME activity would be reduced even without induction, but maybe this would warrant testing whether PMEI3 is actually overexpressed and/or whether PME activity is decreased. On a similar note, the PMEI5ox plants do not appear to show the typical phenotype described for this line. I personally don't think these lines are necessary to support the study. Short-term interference with PME activity (such as with EGCG) might be more meaningful than life-long PMEI overexpression, in light of the numerous feedback pathways and their associated potential secondary effects. This might also explain why EGCG leads to an increase in pH, as one would expect from decreased PME activity, while PMEI expression (caveats from above apply) apparently does not (Fig 3A-D).

We agree with reviewer #2. The PMEI3ox line from Peaucelle et al., 2008 is ethanolinducible, but we observed a strong phenotype (at seedling and adult stage) without ethanol induction. We performed all experiments (root growth assays and confocal observations) with as well as without induction using ethanol, leading to similar results. We concluded from that, that the line is either leaky or that overexpression of PMEI3 is already induced upon seed sterilisation with ethanol. Accordingly, we did not intend to use the lines as acute inhibition of PME but rather used the lines to genetically confirm our data derived from acute pharmacological inhibition. We do show in Figure 1G that the levels of de-methylesterified pectin is decreased in the PMEI3ox mutant compared to WT seedlings. It is exactly this alteration that we are exploiting to assess the necessity of charged pectin for RALF1 signalling. Since the apoplastic pH in the PMEI3ox line is not altered compared to WT, we can conclude that the observed effect on RALF1 signalling is entirely due to the altered pectin charge.

We would like to note that the PMEI5ox line indeed shows the reported root-bending phenotype when grown on plates. We started to perform RALF application assays in liquid medium, because EGCG does not show activity on MS plates. Moreover, it allows us to perform the assays with low amounts of synthetic peptides. The seedling images in our root growth assay might be hence misleading since the assay was done in liquid MS medium and the seedlings were carefully straightened on MS plates before imaging. This transfer makes it difficult to observe the root-bending or -curling phenotype, which is typical for PMEI5ox.

At least at first sight, the observation that OGs are able to titrate RALF from pectin binding seems at odds with the idea of cooperative binding with low affinity, leading to high avidity oligomers. Perhaps the can provide a speculative conceptual model of these interactions?

We added a high concentration of OGs in the media and observed a strong repression of RALF1 activity at the root surface. We assume the OGs form oligomers with RALF peptides in the media, preventing them from penetrating the roots.

I could not find a description of the OG treatment/titration experiments, but I think it would be important to understand how these were performed with respect to OG concentration, timing of the application, etc.

Thank you for pointing this out. The description of the OG RALF titration is added in the methods section.

**Reviewer #2 (Recommendations for the Authors):**
Page 3: „and can bind to extracellular pectin" Liu et al, 2024 should maybe also be cited here.

Amended.

I am not so sure about the use of "conceptualizing" in the last sentence of the abstract and elsewhere in the manuscript.I would suggest adding a few sentences that describe and differentiate what this study and other recently published works (e.g. Dünser, Liu, Mossou, Lin) have revealed about the pectin association of RALFs, LRXs, and FER to help the non-expert reader to navigate this increasingly complex area. May also be worth mentioning that the previously described pectin sensing function of FER is physically separated from the RALF binding domain (Gronnier et al., 2022)

Thank you for your constructive comments. We followed your suggestions and further improved the discussion in the revised version of our manuscript.

**Reviewer #3 (Recommendations for the Authors):**
(1) The authors claim that pectin is something like an extracellular signaling scaffold. In other fields, signalling scaffold refers to proteins that tether the signalling components and regulate/are involved in the signal transduction. Here, pectin is a cell wall structural component whose molecular status is sensed and perceived rather than a functional signaling component. To me, it is FERONIA to be called a signalling scaffold in this case. However, this is my view, and the authors may present their concept.

RALF peptides as well as FERONIA bind to de-methylesterified pectin, which is essential for its signalling output. Albeit not being a protein, we propose that pectin functions like a scaffold tethering both signalling components and thereby enabling signalling. FERONIA has been indeed also proposed to function as a scaffold when tethering other signalling components.

(2) I have no problem with authors using the more general term pectin instead of homogalacturonan throughout the text. Still, authors should, at some point in the text, specify that by pectin, they mean homogalacturonan; the authors did not analyze other pectic types on binding.

We followed your suggestion.

(3) The authors show that RALF1 binds to OGs with a high avidity. Given the fact that OGs released from homogalacturonan upon pathogen infection are Damage-Associated Molecular Patterns (DAMPs), this opens the possibility that this particular activity of RALF1 might actually function in modulation of immune response. I suggest that authors should not exclude this possibility.

We fully agree to this possibility for FER-dependent signalling.

(4) Are there any indications that a similar mechanism can be extrapolated to other FERONIA homologs, such as THESEUS or HERCULES? Although it is not essential to comment, I think this could enrich the discussion.

This is a highly interesting research question, which we may follow up in our upcoming studies. RALF34, which is considered a ligand for THESEUS, also induced FER internalization, which was also sensitive to PME inhibition. While this requires further investigation, this finding hints at a common mechanism for FER- and THE-dependent RALF peptides.

(5) I suggest using the model scheme currently in the supplement as a main figure to provide an immediate accessible summary of the findings.

Thank you for the suggestion to add the summary scheme in the main figures. We followed your suggestion.